# Therapeutic Drug Monitoring of Second- and Third-Generation Antipsychotic Drugs—Influence of Smoking Behavior and Inflammation on Pharmacokinetics

**DOI:** 10.3390/ph14060514

**Published:** 2021-05-27

**Authors:** Nicole Moschny, Gudrun Hefner, Renate Grohmann, Gabriel Eckermann, Hannah B Maier, Johanna Seifert, Johannes Heck, Flverly Francis, Stefan Bleich, Sermin Toto, Catharina Meissner

**Affiliations:** 1Department of Psychiatry, Social Psychiatry and Psychotherapy, Hannover Medical School, Carl-Neuberg-Str. 1, 30625 Hannover, Germany; maier.hannah@mh-hannover.de (H.B.M.); seifert.johanna@mh-hannover.de (J.S.); francis.flverly@mh-hannover.de (F.F.); bleich.stefan@mh-hannover.de (S.B.); toto.sermin@mh-hannover.de (S.T.); meissner.catharina@mh-hannover.de (C.M.); 2Department of Psychiatry and Psychotherapy, Vitos Clinic for Forensic Psychiatry, Kloster-Eberbach-Str. 4, 65346 Eltville, Germany; gudrun.hefner@vitos-rheingau.de; 3Department of Psychiatry and Psychotherapy, Ludwig Maximilian University of Munich, Nussbaum-Str. 7, 80336 Munich, Germany; renate.grohmann@med.uni-muenchen.de; 4Department of Forensic Psychiatry and Psychotherapy, Hospital Kaufbeuren, Kemnater-Str. 16, 87600 Kaufbeuren, Germany; eckermann.iapkf@online.de; 5Institute for Clinical Pharmacology, Hannover Medical School, Carl-Neuberg-Str. 1, 30625 Hannover, Germany; heck.johannes@mh-hannover.de

**Keywords:** therapeutic drug monitoring, second- and third-generation antipsychotic drugs, inflammation, smoking behavior, intoxication, reference ranges, clozapine, pharmacokinetics, CYP enzyme induction/de-induction, AGNP consensus guidelines

## Abstract

Both inflammation and smoking can influence a drug’s pharmacokinetic properties, i.e., its liberation, absorption, distribution, metabolism, and elimination. Depending on, e.g., pharmacogenetics, these changes may alter treatment response or cause serious adverse drug reactions and are thus of clinical relevance. Antipsychotic drugs, used in the treatment of psychosis and schizophrenia, should be closely monitored due to multiple factors (e.g., the narrow therapeutic window of certain psychotropic drugs, the chronicity of most mental illnesses, and the common occurrence of polypharmacotherapy in psychiatry). Therapeutic drug monitoring (TDM) aids with drug titration by enabling the quantification of patients’ drug levels. Recommendations on the use of TDM during treatment with psychotropic drugs are presented in the Consensus Guidelines for Therapeutic Drug Monitoring in Neuropsychopharmacology; however, data on antipsychotic drug levels during inflammation or after changes in smoking behavior—both clinically relevant in psychiatry—that can aid clinical decision making are sparse. The following narrative review provides an overview of relevant literature regarding TDM in psychiatry, particularly in the context of second- and third-generation antipsychotic drugs, inflammation, and smoking behavior. It aims to spread awareness regarding TDM (most pronouncedly of clozapine and olanzapine) as a tool to optimize drug safety and provide patient-tailored treatment.

## 1. Introduction

“Dosis sola facit venenum” (“The dose makes the poison”)—as stated by Paracelsus in the 16th century—underlines the relevance of appropriate drug dosing to avoid serious adverse drug reactions (ADRs). Therapeutic drug monitoring (TDM) aids with drug titration as it enables the quantification of patients’ drug levels during treatment. In psychiatry, TDM is of particular relevance for several reasons [1]: (1) the complex (and partially overlapping) symptomatology, heterogeneity, and chronicity of several psychiatric illnesses [2,3,4,5], (2) the vulnerability of mentally ill persons due to cognitive decline or advanced age [6,7], (3) the relatively frequent reduced adherence or increased risk of substance abuse (i.e., overdosing) [5,8]. Further contributing to the necessity of TDM is the narrow therapeutic window of certain psychotropic drugs, their relatively long half-life or that of their (partially) active metabolites [9], the high incidence of polypharmacy (i.e., the simultaneous intake of ≥5 drugs [10]) including self-medication, and thus the elevated risk for drug-drug interactions [11,12]. Moreover, it is estimated that 15–30% of variability in drug response is due to genetic polymorphisms in CYP isoenzymes [13]. In particular, schizophrenia, a complex and heterogeneous mental disorder, which is often difficult to treat, is associated with a higher risk of polypharmacotherapy and trial-and-error treatment choices [5,14,15,16]. Antipsychotic drugs, used to treat psychosis and thus schizophrenia (among other mental disorders), are further reported to show a high interindividual variability in drug serum concentration [17,18,19] and cause a vast array of serious ADRs (such as agranulocytosis, extrapyramidal motor symptoms, or malignant neuroleptic syndrome), justifying consequent drug monitoring [20,21,22,23,24,25,26].

Drug concentrations should be assessed regularly, under consideration of various aspects such as sudden changes in patients’ health or lifestyle. Repeated TDM is necessary, as a drug’s concentration, or more precisely its liberation, absorption, distribution, metabolism, and excretion (‘LADME’), can be modulated by many factors. In this context, inflammation and smoking behavior have been reported to influence activity of cytochrome-P450 (CYP) isoenzymes (the humans’ main metabolizing system for xenobiotic substances), efficacy of drug transporters, and drug binding to plasma proteins (see Section 2.2 and Section 3.2). Inhibition or induction of drug metabolism has diverse consequences depending, for instance, on a drug’s properties [27], pharmacogenetics (i.e., single nucleotide polymorphisms (SNPs) affecting the genes encoding CYP enzymes) [28,29,30,31], the simultaneous intake of other substances, or diet [32,33,34]. Changes in LADME of antipsychotic drugs in the context of smoking and inflammation are particularly relevant, as inflammation-related ADRs such as hepatitis, pancreatitis, and myocarditis [35] have been reported after use of, e.g., clozapine [36]. Clozapine is a so-called p-mGPCR antagonist (=antagonist at multiple G-protein-coupled receptors with pleiotropic effects), and a second-generation (SGA) or atypical antipsychotic drug. It is further associated with agranulocytosis (i.e., a decline of the absolute neutrophil count below 500/μL) resulting in a higher susceptibility to viral or bacterial infections [36].

Recommendations regarding the use of TDM in psychiatry (suggesting indications, level of recommendation, and therapeutic reference ranges (Table 1), with the latter being associated with the lowest risk of ADRs and toxicity, but the highest chance of therapeutic success) are published by the Therapeutic Drug Monitoring task force of the Arbeitsgemeinschaft für Neuropsychopharmakologie und Pharmakopsychiatrie (AGNP, Association for Neuropsychopharmacology and Pharmacopsychiatry) in 2004 [37], 2011 (first update [38]), and 2017 (latest version [9,39]). 

Although recommended by AGNP, TDM is still underused in psychiatry and not regularly considered after changes in patients’ smoking behavior or during an active infectious disease. Moreover, as schizophrenia and its comorbidities are themselves associated with inflammation (see Section 3.1), successful therapy might also alter patients’ immunological profiles. In fact, antipsychotic drugs have been associated with immunoregulatory properties. Altogether, these factors challenge the prediction of the dose-response relationship of psychotropic drugs and highlight the necessity of regular TDM in clinical routine [41], as the influence of both factors on drug levels is often insufficiently addressed by pharmaceutical companies during drug development [27]. If these factors do find consideration, analyses are often based on in vitro models with limited relevance for human metabolism [15]. The following narrative review sheds light on this topic by summarizing the current literature regarding TDM in psychiatry, particularly in the context of antipsychotic drugs, inflammation, and smoking behavior. Primary focus is the use of second- and third-generation (atypical) antipsychotic drugs, as this class of pharmaceuticals is widely used not only in the treatment of schizophrenia but also in the treatment of psychotic symptoms in other illnesses, such as bipolar disorder and depression. This review aims to spread awareness regarding TDM, as data about the influence of several intrinsic and extrinsic factors on antipsychotic drug levels (but also on reliable therapeutic reference ranges) are mostly lacking. Our article may encourage the publication of case reports/series or conduction of pharmacokinetic studies (whether prospective or retrospective using already existing TDM databases) to provide more information about the use of antipsychotic drugs during, for example, states of infection or changed smoking habits. We hope to contribute to the improvement of drug safety and optimization of patient-tailored treatment.

## 2. Influence of Smoking Behavior on Pharmacokinetics

### 2.1. Smoking Behavior of Psychiatric Patients

Smoking affects the metabolism of many psychotropic drugs; therefore, smoking behavior of patients taking psychotropic drugs can influence the required dose of a drug [42]. The proportion of patients with a mental illness who smoke is two to three times higher than within the general population in Western industrialized nations [43,44]: whereas smoking rates of the general population range from 23% (Sweden) to 37% (Denmark), smoking prevalence among patients of all psychiatric diagnosis groups is estimated at 35–54% [45]. Various epidemiological studies have shown a link between certain psychiatric disorders and an increased rate of smoking among affected individuals. Prevalence of smokers in both the inpatient and outpatient setting in several consistent studies are well above the expected value [32,45] among patients with disorders from the schizophrenic spectrum [46] or affective disorders [47], and especially among patients suffering from addiction [48]. Stockings et al. found that the proportion of smokers in patients with anxiety and personality disorders is 36–39%, in patients with mood disorders 36–49%, and in patients with schizophrenic disorders >60%. Within the psychiatric inpatient setting, the proportion of patients who smoke is as high as 80% [49]. Patients suffering from schizophrenia are particularly vulnerable. These patients are at higher risk of starting smoking than healthy controls while at the same time smoking cessation rates are markedly lower [50]. In this respect, studies have shown that it is more difficult for people with mental illnesses to quit smoking than for people who do not suffer from psychiatric disorders [44]. 

Patients with depressive disorders also smoke more frequently than the general population. Several genetic studies have found that there is a correlation between smoking status and major depressive disorder (MDD) [51,52,53,54,55]. A recent genome-wide association study on tobacco use revealed a negative genetic correlation for MDD with age at smoking initiation and a positive genetic correlation between MDD and smoking initiation in general as well as cigarettes per day [53]. Additionally, smoking contributes to modifications of the oral, lung, and gut microbiome, potentially causing various diseases [56,57,58,59]. The bidirectional cross-talk between the microbiome and the brain (i.e., the gut–brain axis) has recently received increased attention regarding its role in the regulation of behavior and has been considered as a possible biological basis of psychiatric disorders. Several recent clinical studies have linked the microbiome with neuropsychiatric conditions such as depression, schizophrenia, and bipolar disorder [60,61,62,63]. The microbiota of patients with schizophrenia was found to be less diverse in terms of α-diversity (i.e., regarding the actual number of total species) though its composition showed great regional differences within the gut [64,65]. In this sense, healthy subjects and patients with schizophrenia were shown to differ in their β-diversity, which reflects the differences in gut microbiota composition between distinct intestinal sections [66]. 

A change in a patient’s smoking behavior is to be expected especially before, during, and after inpatient care. For example, many patients with a schizophrenic disorder have a higher use of tobacco when suffering from productive psychotic symptoms. During inpatient treatment, smoking behavior often changes, as a result of extrinsic factors such as admission to a secure unit or the use of physical restraint (if medically indicated). In addition, many hospitals pursue a “non-smoking policy” [42,67]. It is important to note that patients are highly likely to return to their original smoking behavior after an inpatient stay, so that their medication also needs to be monitored after discharge from hospital [42].

### 2.2. Effect of Smoking on Pharmacokinetics

#### 2.2.1. The Influence of Smoking on CYP Enzymes

Smoking influences the pharmacokinetic properties of multiple psychotropic drugs. In this regard several ingredients in tobacco smoke (especially polycyclic aromatic hydrocarbons) are of clinical and pharmacological importance. Chemical toxins are potent inducers of hepatic CYP isoenzymes in humans [68,69,70,71,72]. The isoenzymes CYP1A1 (tissue distribution: adipose tissue, lung, oral and pharyngeal mucosa, placenta, uroepithelium, fetal lung, fetal liver) [73,74,75,76,77,78,79,80,81,82,83], CYP1A2 (liver) [84,85,86], CYP1B1 (adipose tissue, lung, oral mucosa, placenta, whole-blood cells, fetal lung) [73,74,76,77,81,83,87,88,89], and CYP2S1 (bronchoalveolar macrophages) [90] are induced by polycyclic aromatic hydrocarbons in cigarette smoke through activation of the aryl hydrocarbon receptor (AhR). In addition, cigarette smoke contains benzene derivatives like styrene and toluene, which induce CYP2E1 (liver, bronchial epithelium) [91,92]. Not all of these isoenzymes are relevant in pharmacokinetics. The primary pharmacokinetic interactions with smoking affect drugs that are substrates of CYP1A2 (Table 1), such as clozapine, fluvoxamine, and olanzapine [70]. The maximum induction of CYP1A2 is reached after the consumption of 10 cigarettes. Therefore, the consumption of ten or thirty cigarettes a day does not induce difference in drug serum concentrations of CYP1A2 substrates, but the consumption of five or ten cigarettes makes a clinically relevant difference in drug serum concentration and therefore in clinical decision making [93,94].

Reduced tobacco consumption or smoking cessation can lead to a reduced activity of CYP1A2 or other CYP isoenzymes potentially causing a clinically relevant increase in the drug level of CYP1A2 substrates [70]. The de-induction of CYP1A2 occurs with a temporal latency of about three days up to one week before a clinically relevant increase in the drug level of CYP1A2 substrates is to be expected [69,95]. An increase in the plasma level of a psychotropic drug may lead to an increased incidence of serious ADRs as well as possible drug intoxication. Patients using drugs metabolized via CYP1A2 should be made aware of this in order to initiate dose adjustment if necessary [96]. Electronic cigarettes do not affect drug metabolism in this manner, as they do not contain polycyclic aromatic hydrocarbons and benzene derivatives. Thus, a switch from cigarettes (or tobacco products in general) to electronic cigarettes may result in decreased drug metabolism—the same holds true for nicotine patches. In this case, a de-induction of CYP1A2 activity is to be anticipated [69,97]. 

#### 2.2.2. Smoking-Associated Pharmacokinetic Changes Due to Other Mechanisms

While nicotine does not interfere with CYP enzymes, nicotine contained in tobacco smoke can influence the pharmacokinetic properties of drugs in another way. A study by Tega et al. suggests that the H+/organic cation antiporter contributes to the transport of nicotine across the blood–brain barrier, which also interacts with several central nervous system (CNS) drugs. Inhibition studies in TR-BBB13 cells showed that nicotine uptake was significantly reduced by CNS drugs such as antidepressant drugs, drugs used in the treatment of Alzheimer’s disease, and drugs used in the treatment of Parkinson’s disease, suggesting that the nicotine transport system may recognize these drugs [98]. Further research is needed to improve the understanding of the interaction between nicotine and CNS drugs.

In addition, smoking cigarettes reduces blood circulation due to vasoconstriction and thus diminishes the oxygen supply to certain tissues including the liver [99,100,101]. Oxygen is a mandatory cofactor of CYP isoenzymes. By reducing the oxygen supply to the liver as a result of smoking, the CYP isoenzymes cannot function properly, thereby curbing drug metabolism [102]. To date, however, no statements can be made about the clinical relevance of this effect.

More recently, the field of epigenetics has become increasingly important, with deoxyribonucleic acid (DNA) methylation being the best-characterized epigenetic modification. In some studies, epigenetic variation induced by environmental exposure has been hypothesized as another mechanism by which non-genetic factors can affect risk for neuropsychiatric disorders including schizophrenia as well as efficacy of antipsychotic medication [103]. Hannon et al. have shown that there are differences in DNA methylation patterns in patients with schizophrenia, taking into account patients’ smoking status as well as drug treatment with clozapine or other antipsychotic drugs [104]. Much more research is required in order to fully understand the influence of smoking behavior on DNA methylation and possibly also on the efficacy of antipsychotic drugs, especially in treatment-resistant schizophrenia.

As mentioned in Section 2.1, smoking affects the microbiome. Clinicians treating patients with psychotic illness have recently begun to realize that gut bacteria play a key role in the effectiveness of antipsychotic medication and its tolerability and toxicity [105]. Drugs metabolized by gut bacteria may result in different drug metabolites potentially enhancing or diminishing a drug’s efficacy [106]. Besides, increases in intestinal permeability caused by a microbiota-associated leaky gut were suggested to enhance drug absorption [65]. Changes in gut bacteria by smoking might thus alter a drug’s pharmacokinetics but may also increase the risk of ADRs as decreases in gut microbiota diversity were associated with metabolic syndrome, obesity, or coronary vascular diseases [107]. 

In order to avoid serious ADRs (see Table 2), it is important to be aware of a patient’s smoking status, to inform the patient about possible smoking-induced changes in drug metabolism, and, if necessary, to monitor drug levels via TDM. The influence of smoking on pharmacokinetics and pharmacodynamics of selected psychotropic drugs, which are mainly used in the treatment of schizophrenia, is explained in Section 6.

## 3. Inflammation—Its Relevance in Psychiatry and Pharmacokinetics

### 3.1. The Role of Inflammation in Psychiatric Patients

#### 3.1.1. Mental Illness and the Immune System

Inflammatory processes are relevant in schizophrenia, not only during the ongoing coronavirus disease 2019 (COVID-19) pandemic. Quite the contrary, inflammation has been considered as a state marker for acute or first-episode psychosis [111,112,113,114], but both immune system activation and immunosuppression are prevalent in affected subjects [115]. The latter manifests in (1) an increased susceptibility to (viral) infections [116] with influenza and pneumonia being frequent causes of death in schizophrenic patients [117], (2) a declined mitogen-induced proliferation of lymphocytes [118], and (3) diminished numbers of total T cells [119]. On the other hand, immune system activation is indicated by increases in pro-inflammatory cytokine plasma levels such as interleukin (IL)-1β, IL-6, soluble IL-2 receptor (sIL-2R), and tumor necrosis factor alpha (TNFα) [113,120,121,122,123,124], in immunoglobulins [116], and, lastly, in certain cell subtypes of the adaptive immune system (e.g., monocytes and neutrophils) [125,126]. The specific immunological phenotype depends on various factors, with treatment resistance being associated with a more pronounced pro-inflammatory status [115]. The duration, current phase (whether prodromal, acute exacerbation, or relapse, for instance), and symptomatology of the illness (e.g., the extent of sleep dysregulation or negative symptoms and thus social isolation), or the resulting behavior of schizophrenic patients (smoking is reported to induce inflammation) are further contributors potentially modulating the immunological phenotype [115,122,127,128,129,130,131,132]. Evidence suggests that the CNS is also immunologically challenged—studies have shown increased numbers of ionized calcium-binding adapter molecule-1 positive (Iba1+) microglial cells (the main actors regarding the brain’s immune system) in the corpus callosum of myelin-basic protein heterozygous (MBP+/−) shiverer mice, a rodent model of schizophrenia [133]. Further evidence derives from various studies, reporting raised (R)-[11C]PK11195 binding (a positron emission tomography tracer measuring the abundance of 18 kDA translocator protein and thus microglial activity) [134] or elevated inflammatory markers in the cerebrospinal fluid of patients with schizophrenia [135]. Nevertheless, this topic is still a matter of debate as conflicting data (showing comparable results to healthy controls or highlighting the discrepancies of the methodology used in the above cited studies) have also been found [136,137]. Immunological dysfunction in the CNS and in the periphery resulting in inflammation may cause neurodegeneration and aberrant neurotransmission [138,139,140,141,142,143,144], both of which are proposed to play a role in the etiology and/or pathophysiology of schizophrenia. This also holds true for abnormalities in microbiota that, as delineated above, are thought to play an important role in schizophrenia. Altered composition of the microbiota might lead to changed fermentation products that influence vagal and endocrine pathways, epithelial integrity, immune system functioning, and thus the brain [145]. Particularly in the case of a leaky-gut-syndrome, which is associated with changes in gut microbiota composition and allows lipopolysaccharides (LPS) to enter the blood stream, chronic inflammatory stress may challenge brain homeostasis [146]. 

#### 3.1.2. Psychotropic Drugs and the Immune System

Since immune processes have been considered to play a fundamental role in the development of schizophrenia, immunoregulatory properties of antipsychotic drugs have recently become a focus of research interest. Antipsychotic drugs have been proposed mainly to cause immunological suppression as assumed by their capacity to stimulate IL-10 or IL-1RA, and to reduce pro-inflammatory TNFα, IL-6, or interferon gamma (IFNγ) cytokine production [111,115]. The effect has been stated to rely on the drug type, its concentration, intake duration, and environmental factors [111]. Short-term medication with clozapine, for example, induced a rather pro-inflammatory response [115]; the same holds true for chlorpromazine and haloperidol at lower concentrations as demonstrated in an in vitro model of LPS-stimulated primary rat mixed glial cells [111]. 

Further highlighting the relevance of inflammation in psychosis, antipsychotic drugs occasionally cause severe inflammatory conditions such as myocarditis, pericarditis, or metabolic syndrome [36]. This effect was suggested to be (at least in part) mediated by the gut microbiota, as (1) antipsychotic drugs modulate its diversity/composition and (2) changes in microbiota were found to impact weight gain and the risk for metabolic diseases by altering gene expression and inflammatory and metabolic pathways [65,147]. Moreover, not only antipsychotic drugs but also medication used in the treatment of several critical or frequent comorbidities of schizophrenia (such as depression, post-traumatic stress disorder, or cardiovascular diseases) [5,148], are associated with immunomodulatory effects. Depression, for instance, has not only been stated to be partially caused by an aberrant immune system (‘inflammatory hypothesis of depression’), but antidepressant drugs also have an impact on cytokine expression [149]. Lastly, clinical studies suggest that anti-inflammatory agents (e.g., acetylsalicylic acid, minocycline, celecoxib, or infliximab) exert beneficial effects in the treatment of both schizophrenia and depression [112,150,151,152]. 

While it is necessary to bear in mind the heterogeneity of patients with schizophrenia regarding immune system irregularities [2,3,4,153], TDM is still of paramount importance, as acute infections may affect all patients. Continuous TDM might be particularly relevant as inflammatory changes are highly dynamic and dependent on the current phase or duration of the illness, its comorbidities, and the type, duration, and concentration of the drug. These variables, potentially resulting in different immunological phenotypes impacting drug absorption, metabolism, excretion, or pharmacodynamics differently, may contribute to the unpredictability of dose–response relationships [41], especially in the case of polypharmacotherapy. The complex interplay between inflammatory factors, drug-metabolizing enzymes, transporters, or drug-binding proteins, and its relevance regarding particular psychotropic drugs, is described in Section 3.2 and Section 7. 

### 3.2. Inflammation-Associated Pharmacokinetic Changes

#### 3.2.1. Physiology of Inflammation

Inflammation aims at restoring tissue homeostasis after being challenged by extrinsic or intrinsic insults. The release of inflammation-related substances (such as prostaglandins, leukotrienes, cytokines (including chemokines), vasoactive amines, and plasma endopeptidases, among other factors) [31] is induced by damage- or pathogen-associated molecular patterns (i.e., host-derived endogenous signaling molecules or pathogenic fragments, respectively) that bind to pattern-recognition receptors located intracellularly or at the membrane of several immune cells. In this context, particularly immune cells belonging to the innate immune system (macrophages, monocytes, neutrophils, and dendritic cells) are well-equipped with toll-like receptors (TLRs), which detect tissue injury or extraneous particles. Binding to TLRs results in an activation of the nuclear factor kappa-light-chain-enhancer of activated B cells (NF-κB), a central transcription factor family involved in pro-inflammatory cytokine production [154,155,156,157]. This cascade (also named acute-phase reaction, APR) not only serves in the recruitment of peripheral immune cells to the site of tissue damage [158,159,160] enabling, for example, enhanced phagocytosis and thus tissue repair [161,162,163], but also leads to changes in drug metabolism [27,31,41,157].

#### 3.2.2. Possible Mechanisms of Inflammation-Induced Pharmacokinetic Changes

APR-associated changes in drug metabolism are thought to be induced by altered CYP enzyme activity and do not only hold true for acute infections (as the expression ‘APR’ might suggest), but also for chronic inflammatory diseases, depending on the amount and type of secreted inflammatory stimuli. In this context, pro-inflammatory cytokines (e.g., IL-1, IL-2, IL-6, TNFα/β, or IFNγ) may modulate hepatic protein expression via NF-κB, as cytokine-induced NF-κB has been found to inhibit the heterodimerization of nuclear receptors known to play a key role in the regulation of genes encoding for drug-metabolizing enzymes, such as of the retinoid X receptor (RXR)-α, the constitutive androstane receptor (CAR), the peroxisome proliferator activated receptor (PPAR), and the pregnane X receptor (PXR) [157,164,165]. This evidence is supported by various studies. Tanner et al. [166] reported that IL-6 exhibited a pronounced negative effect on PXR, CAR, and PPARα (but not on AhR-dependent activation of drug metabolism), which rather results from blocked transactivation than from diminished nuclear receptor-encoding messenger ribonucleic acid (mRNA) transcription. Inhibition as a consequence of failed mRNA transcription was described by Klein et al. [167] who found suppressed mRNA expression of CAR and PXR after stimulating human hepatocarcinoma cells (= HepaRG cells) with IL-6. Post-transcriptional processes, such as nitric oxide (NO)-dependent proteasome degradation of drug-metabolizing enzymes, were also suggested to be of importance [157]. For example, downregulation of CYP enzyme activity has been prevented by the use of proteasome or NO synthase inhibitors [168,169,170]. On the other hand, clearance of CYP hemeproteins (but not mRNA) was induced within 3 h after applying a NO donor to HeLa cells [171]. In addition to these possible mechanisms, epigenetics was ascribed a role in drug-metabolizing enzyme inhibition upon inflammation. In this context, DNA methylation of CYP11A1 has been found to be associated with C-reactive protein (CRP) levels measured in patients with bipolar mania [172], and DNA methylation levels in promoter regions of several CYP enzymes to inversely correlate with the enzymes’ hepatic expression [173]. For more details regarding the mechanisms of drug-metabolizing enzyme inhibition by inflammatory substances, we refer to the review by Stanke-Labesque et al. [157]. 

#### 3.2.3. Effects of Inflammation on CYP Isoenzymes

IL-6, TNFα, and IL-1 were proposed to be the main effectors in the inhibition of drug-metabolizing enzymes, though their effect differs in terms of strength and preference for specific CYP isoforms [31,174]. The liver has been stated to be a major target of IL-6, as its receptors (IL-6R and gp130) are expressed by hepatocytes [175,176]. Several reports confirm its inhibitory effect showing CYP1A2, CYP2B6, CYP2C8, CYP2C9, CYP2C19, and CYP3A4 activity to be decreased after stimulating HepaRG cells with IL-6 for 48–72 h [167]. Additional evidence was provided by Dickmann et al. who demonstrated suppressed CYP3A4 activity in primary human hepatocytes after exposure with higher IL-6 concentrations [177]. IL-6 was further shown to suppress CYP1A2, CYP2B6, and CYP3A4 mRNA levels [178] and to be associated with inhibited CYP3A4, CYP1A2, and CYP2C19 functioning (but not CYP2E or CYP2D6) in patients with cancer, congestive heart failure, or post-surgery [41,179,180,181]. Administration of tocilizumab (an anti-IL-6 receptor antibody used in the treatment of rheumatoid arthritis), increased the activity of CYP3A4 and thus simvastatin clearance [182]; however, no effect was prevalent regarding CYP2C19 and CYP2D6 as measured by the elimination of omeprazole and dextromethorphan, respectively [27,183]. Regarding IL-1, in vitro studies demonstrate a marked downregulation of CYP2C8 and CYP3A4 mRNA, but not of CYP2C9 or CYP2C19 [184]. Klein et al., who stimulated HepaRG cells with cytokines, confirmed these findings, as CYP activity decreased drastically after exposing HepaRG to IL-1 or TNFα (instead of IL-6), though in this case, all investigated CYP isoenzymes (CYP1A2, CYP2C8, CYP2C9, CYP2C19, CYP3A4) were found to be affected [167]. With regard to TNFα, not only Klein et al. [167] but also the above-mentioned article by Frye et al. [179] demonstrated its inhibitory properties showing CYP2C19 activity to negatively correlate with TNFα levels. Another example of this effect is that antagonization of soluble TNFα using biologicals (i.e., XPro1595, a dominant-negative form of TNFα) prevents downregulation of CYP3A11 and CYP3A25 mRNA—in fact, a reversed effect, i.e., an increased CYP activity was observed—but not of CYP2B10, CYP4A10, or CYP4A14 [185]. Besides IL-1, IL-6, and TNFα, other cytokines have received far less attention. In this context, studies showed suppressive effects for IL-10 (reduction of CYP3A activity with no influence on CYP1A2, CYP2C9, or CYP2D6) [186], but not for IL-12 or IL-23 [31,187]. 

Suppression of drug-metabolizing CYP enzymes may result in so-called pheno-conversion, meaning the transient change of genotypic extensive metabolizers into phenotypic poor metabolizers. In this context, genetic variants have mostly been described for CYP2D6, CYP2C19, and thiopurine methyltransferase (TPMT; another important drug-metabolizing enzyme), resulting in four (in the case of TPMT only three) different genotype-based subpopulations of the respective enzymes. These subpopulations have differing kinetics leading to ultra-rapid, extensive, poor, or intermediate drug metabolism in affected subjects [29,30,31]. Pheno-conversion is of great relevance, as the phenotype of metabolism (meaning the ‘true’ kinetics, involving the genotype as well as other influencing factors, such as inflammation) rather than the genotype alone was reported to be a better predictor of treatment response in the case of CYP2D6 substrate antidepressant drugs [188]. Studies based solely on CYP genotypes have been proposed to lead to conflicting results, as the phenotype is not taken into account [31,189]. 

#### 3.2.4. Effect of Inflammation on Other Drug-Metabolizing Proteins and Transporters

CYP enzymes (that belong to the class of phase I drug-metabolizing enzymes) are substantial effectors in drug metabolism: CYP3A4, for instance, metabolizes almost 50% of all currently available drugs [190,191,192]. However, other relevant proteins must also be considered: uridine diphosphate-glucuronosyltransferases (UDP-GTs), sulfotransferases, glutathione S-transferases (GSTs), N-acyltransferases, and methyltransferases. All of these enzymes are categorized as so-called phase II conjugative enzymes, as they add particular moieties to drugs to modulate their lipophilic/hydrophilic properties [27]. As in the case of inflammatory substances other than IL-1, IL-6, and TNFα, phase II enzymes have received much less attention than CYP enzymes. The same holds true for cellular transporters, such as adenosine triphosphate (ATP)-binding cassette transporters (ABCs; drug efflux) and the solute-carrier (SLC) family of uptake transporters, which determine bioavailability, distribution, and elimination of a drug depending on their location [193]. Similar effects as for CYP enzymes have been reported under inflammatory conditions for both phase II enzymes and transporters: in the above-mentioned study by Klein et al., stimulation of HepaRG and primary human hepatocytes with IL-6 led to a similar decline of UDP-GTs, ABCs, SLCs, and GSTs than of CYP enzymes [167]. These findings have been corroborated by other investigators in patients with ulcerative colitis [194] and liver disease [195] or in rodents infected with pathogens or stimulated with LPS A, which is a membrane fragment of gram-negative bacteria [157]. In animal studies, pathogenic infection led to decreases in UDP-GT1A1 and UDP-GT2B mRNA expression [196] or hepatic SLC22A4, SLCO1A1, SLC1A4, SLC02B1, and ABCC6, whereas declines of ABCC2, ABCC3, and ABCC4 were found to be strain-dependent [197]. Supporting evidence is provided by Piquette-Miller et al. [198] reporting turpentine- or LPS-induced acute inflammation in rats to suppress hepatic protein expression of P-glycoprotein (P-gp or ABCB1)—an essential transporter in drug efflux also in the brain. Increased brain penetrance of opioids (such as loperamide, fentanyl, and morphine) as a result of inflammation was confirmed by Puig et al. using mice injected with croton oil [199].

#### 3.2.5. Conflicting Data on the Effects of Inflammation on Pharmacokinetics

Overall, these data indicate a pronounced down-regulation of drug-metabolizing enzymes (phase I as well as phase II) and drug transporters during states of inflammation. It is thought that this shut-down aims to redistribute the necessary amino acids, lipids, and nucleotides, to enable production of APR proteins such as α1-acid glycoprotein, CRP, and α1-antitrypsin [193]. However, other studies have partially demonstrated enhanced activity of CYP enzymes and transporters upon inflammatory stimulation or conditions. In this context, Klein et al. reported CYP2E1 to be increased by almost twofold after stimulation of primary human hepatocytes (PHH) with IL-6 for 24 h [167]. Interestingly, HepaRG cells reacted differently to IL-6 exposure, as CYP2E1 was found to be slightly (but not significantly) decreased. Divergence between both in vitro cell types were also found in relation to all other CYP enzymes analyzed (CYP1A2, CYP2B6, CYP2C8, CYP2C9, CYP2C19, and CYP3A4), as only HepaRG but not PHH showed a significant down-regulation, further impeding interpretation of these findings. Regarding validity, one has to keep the expression level of these enzymes in mind; the lacking induction of CYP2E1 in HepaRG cells, for example, was ascribed to its scarce presence in these cells. This is also of particular relevance for CYP2D6, an important isoform metabolizing 20% of all CYP-dependent drugs [31], which seems not to be expressed by HepaRG cells at all [167]. Further discrepancies were reported by Molanaei et al. whose initial results (a down-regulation of CYP3A4 activity upon inflammation, i.e., elevated high-sensitive CRP) were not significant after multivariate regression analysis [190]. Lastly, Merrell et al. reported mRNA expression of SLCO3A1 and ABCB1B to be raised in a strain-dependent manner, as opposed to other transporters that were found to be blocked in rodents [197]. Findings in patients are based on a wide variety of diseases (e.g., cancer, kidney diseases, liver diseases), further complicating a comparison of results. Moreover, oroso-mucoid (synonym: α1-acid glycoprotein), an inflammatory marker elevated during immunological challenges, was found to bind to drugs and might thus lead to the decrease of their unbound fraction [190,200]. As a consequence, drug concentrations may only appear to be increased due to inhibited drug metabolism, but only the free fraction is available for a drug’s distribution and clearance [27]. However, this may be less relevant for drugs that only have very poor distribution due to their biochemical properties, at least in comparison to drugs with a high volume of distribution [27,201]. As albumin levels (another drug-binding protein with high affinity to smaller molecules and acidic drugs [202,203]) were reported to be mostly diminished during inflammation [193], the question rises as to whether the increase in oroso-mucoid is of clinical relevance. 

Overall, these data underline the complexity of pharmacokinetics during inflammation, making a clear judgment about the effect on a specific drug’s absorption, distribution, metabolism, and clearance difficult. Pharmacokinetics highly depend on the drug’s chemical properties, the patient’s characteristics (e.g., volume of distribution, pregnancy, illnesses, genotype), the inhibited enzyme/transporter, the pharmacodynamics of the drug’s metabolite(s), and the patient’s co-medication. Nevertheless, these findings clearly demonstrate that awareness should be raised regarding the re-evaluation of drug dosage during inflammation as this might modulate drugs’ plasma concentrations and increase the risk for ADRs or subtherapeutic concentrations. 

## 4. Antipsychotic Drugs 

Antipsychotic drugs are primarily used in the symptomatic treatment of illnesses from the schizophrenic spectrum as well as for psychotic symptoms in the context of other disorders (including affective or organic disorders) and mania. Antipsychotic drugs decrease psychomotor agitation and reduce affective tension, anxiety, formal thought disorders, and delusional perceptions. They are also used in the treatment of children and adolescents with manic syndromes and (only applicable to some antipsychotic drugs) as mood stabilizers, for tic disorders, irritability, agitation, and aggressive behavior, for example in the context of autistic or other developmental disorders, as well as in patients with impulsive-expansive behavior disorders with or without intellectual disabilities [204].

Antipsychotic drugs can be broadly divided into “conventional” and “atypical” antipsychotic drugs. First-generation or conventional antipsychotic (FGAs) drugs are high-affinity antagonists of dopamine D2 receptors. Therefore, they are most effective in the treatment of psychotic (positive) symptoms but have high rates of neurologic ADRs such as extrapyramidal symptoms and tardive dyskinesia [205]. The advent of SGAs (=“atypical” antipsychotic drugs) promised enhanced efficacy and safety [206]. SGAs differ pharmacologically from FGAs in their lower affinity for dopamine D2 receptors and greater affinities for other neuroreceptors, including serotonin (5-hydroxytryptamine 1A, 2A, 2C, 3, 6, and 7) and norepinephrine (α1 and α2 adrenoceptors) [205]. They are also more effective in the treatment of negative symptoms than FGAs, combined with a presumably reduced rate of extrapyramidal side effects.

The classification of antipsychotic drugs as conventional and atypical antipsychotics has been subject to fierce controversy among experts. As a consequence, other possible classifications of antipsychotic drugs have emerged, pertaining, for example, to their chemical structure or their neurobiological mode of action. However, so far none of these classifications has become sufficiently established in clinical routine [207]. 

As of 2004, a new group of antipsychotic drugs became available. These so-called third-generation antipsychotic drugs (TGAs) differ from SGAs due to a partial agonism at the dopamine D2 and D3 receptors [208,209]. Due to the more favorable side-effect profile, SGAs and TGAs are preferred nowadays [210], which is why we will mainly refer to these in the following. For an overview of TDM as well as the pharmacodynamics of SGAs and TGAs, see Table 1, Table 2 and Table A1.

## 5. Therapeutic Drug Monitoring of Pharmaceuticals

As previously indicated in the introduction of the current review, recommendations for implementing TDM in clinical routine come from the Therapeutic Drug Monitoring task force of AGNP, a German interdisciplinary association of experts in neuropsychopharmacology. Their consensus guidelines are based on the assumption of a correlation between drug blood levels, clinical outcome, and ADRs (i.e., chemical-clinical correlations) [15,211]. They are further based on the principle of ‘therapeutic indices’ (the so-called ‘therapeutic window’), meaning a particular blood concentration range that is associated with the highest degree of drug efficacy and lowest risk of ADRs and toxicity [15]. In short (for detailed information we refer to the original publication [9], or the summary of its latest version [39]), their guidelines address theoretical as well as practical aspects of TDM and provide therapeutic reference ranges for 154 neuropsychiatric drugs, with the latter term being officially defined as ‘ranges of drug concentrations in blood that specify a lower limit below which a drug induced therapeutic response is relatively unlikely to occur and an upper limit above which tolerability decreases or above which it is relatively unlikely that therapeutic improvement may be still enhanced’ [9]. These therapeutic reference ranges refer not only to the initially applied chemical substance, but to the active moiety, i.e., the sum of parent compound and its pharmacodynamically active metabolites [39,212]. Moreover, the guidelines give other indicators facilitating personalized drug dosing, such as metabolite to parent compound ratios (reflecting metabolic activity, and—if a drug’s metabolism depends on solely one defined CYP enzyme—the pharmacogenetic phenotype) or laboratory alert levels that when exceeded may correlate with an elevated risk for ADRs/toxicity. In order to facilitate the implementation of TDM, AGNP categorizes relevance of TDM for each drug (strongly recommended—recommended—useful—potentially useful). It further provides factors for the calculation of dose-related drug concentrations (C/D), indicating the minimal expected blood concentration of a defined daily dosage upon steady-state conditions (so-called ‘trough levels’). In steady-state, drug input equals drug elimination and is reached after 4–5 half-lives of the respective drug. Blood for TDM should be drawn at trough levels, i.e., immediately before the first daily dose or—in case of long-acting injectables—next injection. Caution should be exercised with depot or extended-release medication, as pharmacokinetics may vary depending on the galenic formulation. Approximately 90% of psychotropic drugs reach steady-state after one week of treatment, although it should be noted that inducers or inhibitors of metabolism (potentially resulting in drug-drug interactions) should also have reached steady-state conditions [9,39]. Due to relatively long half-lives, steady-state is often reached later with TGAs.

Absence of therapeutic reference ranges restrict the usefulness of TDM [1]. If available, these ranges refer solely to the primary indication of the drug and are not applicable to children and elderly patients. Because many factors may affect a drug’s pharmacokinetics in an individual, the AGNP guidelines should not be seen as the ultimate parameter according to which mandatory action must be taken (if a laboratory alert level has been exceeded, for instance). The treating physician must decide whether to adjust drug dosage by taking into account multiple factors, including age, diagnosis, and current presence of ADRs [39]. In this sense, AGNP itself concludes that ‘neuro-psycho-pharmacotherapy can be best guided by identification of the patient’s individual therapeutic concentrations’ [9]. Despite the delineated shortcomings, there is clear evidence that TDM benefits personalized therapy [1], compensating for its economic costs by decreasing the need for treatment of ADRs and hospitalization.

## 6. Effect of Smoking on Antipsychotic Drug Levels

### 6.1. Effects of Smoking on Drug Levels of Antipsychotic Drugs Metabolized via CYP1A2

Since the polycyclic aromatic hydrocarbons in tobacco smoke mainly affect CYP1A2, most data on smoking behavior and changes in plasma concentration pertain to SGAs such as clozapine and olanzapine, which are CYP1A2 substrates. In general, smokers have lower serum levels of clozapine and olanzapine than non-smokers when treated with the same oral dose [33,67,213,214,215,216,217,218,219]. Published reports document 20–40% lower mean serum clozapine concentrations in smokers (up to 14 cigarettes per day) compared with nonsmokers due to enzyme induction [67]. When a patient stops smoking, dose reduction of clozapine of 30–50% is recommended [213,215]. In a Chinese study of 354 adult psychiatric patients, olanzapine elimination was increased 1.23-fold by smoking [214]; therefore, Tsuda et al. suggest that the dose of olanzapine should be reduced by 30% in non-smokers compared with smokers to obtain an equivalent olanzapine concentration [213]. 

Overall, blood levels are highly variable. In addition to smoking behavior and interindividual differences (e.g., genetic variations like a novel variant in the nuclear factor 1 B-type gene associated with reduced clozapine levels [220]) drug concentrations are also influenced by other factors such as age, gender, origin, and co-medication [214,215,217,218,219]. However, previous research has demonstrated that, after clozapine dosage itself, smoking is the factor that most affects clozapine plasma concentration [67]. 

Hence, a change in smoking behavior of patients who are stable on clozapine or olanzapine is of great clinical importance. A sudden cessation of smoking may result in an intoxication with potentially life-threatening symptoms. It should be noted that the de-induction of CYP1A2 takes place within three days to one week, so that it is already sufficient if the patient does not smoke for three to five days [69,95]. The increase in non-smoking psychiatric facilities in the United States has further contributed to this issue. A study from Oregon State Hospital analyzed clozapine level changes before and after implementation of a hospital-wide non-smoking policy in 11 patients who were on stable clozapine doses. A mean increase in clozapine levels of 71.9% (442.4 ng/mL ± 598.8 ng/mL) occurred upon smoking cessation (*p* < 0.034) from a baseline level of 550.2 ng/mL (± 160.18 ng/mL). One serious ADR (aspiration pneumonia) was associated with a non-smoking serum clozapine level of 3066 ng/mL [67]. Other cases of ADRs as a result of increased plasma levels after smoking cessation include confusion in a patient treated with clozapine and debilitating extrapyramidal motor symptoms in a patient treated with olanzapine [221]. Clinicians should be aware of the potential risks associated with smoking cessation in patients treated with a stable dose of clozapine or olanzapine. Toxicity as a result of recent smoking reduction or cessation may lead to hospital admission [222]. Clinicians should obtain information concerning smoking status, including cessation attempts, when treating psychiatric inpatients. Nonspecific signs and symptoms of elevated clozapine or olanzapine concentrations should be considered in relation to clinical status [222]. 

### 6.2. Effects of Smoking on Antipsychotic Drugs Metabolized by Other CYP Isoenzymes

Since smoking also affects the activity of other CYP isoenzymes, plasma levels of other antipsychotic drugs that are not primarily metabolized by CYP1A2 may be altered. The influence of smoking on blood levels of other antipsychotic drugs appears to be clinically less significant and less data is available than for clozapine and olanzapine. Schoretsanitis et al. have proposed a slight inducing effect of smoking on risperidone metabolism, most likely via CYP3A4. While smokers were treated with a significantly higher daily dosage of risperidone than non-smokers, plasma concentrations of risperidone and its active metabolite 9-hydroxyrisperidone did not differ between the two groups [223]. The opposite effect was found among smokers treated with amisulpride who presented with higher plasma levels of amisulpride than non-smokers, although a significant difference in C/D ratios was not observed [224]. Differences in plasma levels between smokers and non-smokers were also detected among women treated with ziprasidone. This may be in agreement with the fact that the enzyme CYP1A2 is activated by smoking. However, at the current time it is not possible to make any solid conclusions regarding a possible explanation on the differences of smoking effects on dose-normalized ziprasidone concentrations between males and females [225]. Further research is required to ascertain the role of smoking status on plasma amisulpride and ziprasidone concentration.

The plasma levels of other SGAs and TGAs, such as aripiprazole [226], brexpiprazole [227], cariprazine [228], lurasidone [229], quetiapine [230], and sertindole [231], are not affected by cigarette smoking. 

In summary, the interaction between smoking and the plasma level of several SGAs has been intensely studied and is well documented. The vast majority of authors recommend TDM, especially for clozapine and olanzapine, as well as consideration of the patient’s smoking status. However, there is no guideline for clinicians in managing this interaction, only individual suggestions [232]. While the impact of smoking reduction and cessation has been the subject of great scientific interest, further information and guidance on the consequences of increased tobacco consumption is limited. 

## 7. Effect of Inflammation on Antipsychotic Drug Levels 

### 7.1. Inflammation-Induced Toxicity of Antipsychotic Drugs

Literature pertaining to the effect and consequences of inflammation on the absorption, metabolism, and elimination of antipsychotic drugs is scarce. Most currently available data is based on case reports and rarely on retrospective studies investigating larger cohorts. Although data is lacking, the risk of infection-associated clozapine intoxication should not be underestimated: de Leon and Diaz reported a 34-year-old Caucasian male who suffered from myoclonus and increased sedation during a respiratory infection [34]. The patient’s clozapine levels had been monitored 17 times over the course of one year, revealing an elevated C/D ratio of 2.9 upon infection and of 1.0 to 1.6 before and after. Another case report by Bebawi et al. presents a 64-year-old Caucasian woman with general deterioration, drowsiness, neutropenia, and ileus, after developing an acute gastroenteritis/colitis [18]. Determination of the clozapine plasma level revealed a clozapine intoxication (clozapine level 13,156 ng/mL; reference range: 350–600 ng/mL, see Table 1) [9] which was presumably worsened by the patient’s constipation since reduced gastrointestinal motility may increase drug absorption and decline fecal drug elimination [18,27]. Symptoms quickly improved after dose reduction of clozapine and treatment with antibiotics. Leung et al. report further cases of severe clozapine toxicity presumably induced by inflammation and necessitating intensive care [233]. Elevated levels of clozapine (Patient A: 1400 ng/mL, Patient C: 4318 ng/mL) were thought to be caused by an exacerbation of chronic obstructive pulmonary disease (COPD; Patient A), and pneumonia, urinary tract infection, superficial cellulitis, and sepsis (Patient C), respectively. Intoxication became apparent by abrupt mental health changes, myoclonus, hypotension, sialorrhea, and need for assisted ventilation. As in the report by Bebawi et al. mentioned above, adverse effects resolved following dose reduction or pausing of clozapine [18]. The same holds true for a 42-year-old woman presented by Grootendorst-van Mil et al., who experienced severe apraxia and confusion after an infection-associated rise in clozapine levels (1800 ng/mL) [234]. Finally, Tio et al. presented a 46-year-old Caucasian non-smoker suffering from ataxia and tremor during a simultaneous COVID-19- and bacterial respiratory infection. Clozapine trough levels (1814 ng/mL) were thrice as high as during previous lung infections (bronchitis: 616 ng/mL; bacterial pneumonia: 1183 ng/mL). Signs of clozapine intoxication resolved after pausing clozapine for 48 h [235]. 

### 7.2. Drug Levels of Antipsychotic Drugs in Relation to Inflammation

Besides several other case reports demonstrating serious ADRs upon infection-associated clozapine toxicity (see Table 1 in Leung et al. [233]), two retrospective studies confirmed elevated CRP (a non-specific marker of inflammation and tissue damage [40,236]) to positively correlate with the patients’ clozapine levels. To be more precise, Pfuhlmann et al. found that patients with higher blood concentrations of clozapine (>800 ng/mL, n = 27) displayed abnormal CRP levels more often than matched subjects (n = 36) [237]. In addition, these patients’ mean CRP level was higher compared to study participants with lower clozapine concentrations (<600 ng/mL). Hefner et al. analyzed intraindividual drug and CRP levels (>5 mg/L) of 105 patients and revealed a significant association for the C/D ratio of clozapine and risperidone, but only a trend for quetiapine [40]. Instead, a correlation between pathological CRP concentrations and increases of quetiapine’s C/D ratio (n = 166) was shown by Scherf-Clavel et al. [238]: the C/D ratio rose by 0.11 (ng/mL)/(mg/day) with each increase of 5 mg/L CRP. A trend was detected for olanzapine (n = 24) while no association to CRP was found for aripiprazole (n = 30) and risperidone (n = 45). By contrast (at least regarding risperidone), Hefner et al. reported two patients whose dose-related risperidone serum concentrations (including its active metabolite 9-hydroxy-risperidone) increased up to threefold after pneumonia (Patient A) and a non-specific infection (Patient B) [239]. Similar as in the cases of clozapine toxicity, risperidone levels dropped shortly after inflammation subsided. Unfortunately, adverse effects were not recorded in the last two reports, raising the question as to whether raised drug levels were of clinical relevance.

In most of the above-mentioned studies, the metabolic ratios of the parent drug and its metabolite (if measured) were found to be unchanged or within the normal range, although the parent drug’s concentration or its C/D ratio was increased [40,239]. As unaltered metabolic ratios indicate normal CYP enzyme activity [9], this finding is unexpected considering the vast literature emphasizing the inhibitory effect of inflammatory markers (mainly IL-6, IL-1, and TNFα) on CYP metabolism (Section 3.2). In one of the three cases presented by Leung et al. the affected patient presented with fewer clozapine-associated ADRs despite showing the highest drug level of 4318 ng/mL (as compared to Patient A: 1400 ng/mL, and B: 11.30 ng/mL) [233]. As a possible explanation, the authors postulate the high plasma protein-binding capacity of clozapine (approximately 95% of clozapine is bound to plasma proteins [240]) and the acute rise of oroso-mucoid during inflammation. To support this hypothesis, the authors refer to Espnes et al., demonstrating a patient absent of clozapine toxicity despite a level of 2965 ng/mL [241]. As described in Section 3.2, only the free fraction of a drug is distributed and thus exerts physiological effects, be it its desired mechanism of action or undesired adverse reactions. High binding to oroso-mucoid might thus protect patients from symptoms of clozapine intoxication although analytics suggest the drug’s total plasma concentration to be abnormally high [233]. Hefner et al. also conclude that suppression of CYP enzyme metabolism cannot fully explain their findings (i.e., the rise in clozapine and risperidone levels), as—although partially increased—all of their metabolic ratios remained within the normal ranges [40]. Instead, the authors propose inflammation-associated alterations of drug transporters and other drug-metabolizing enzymes to contribute to drug elevation in this particular study.

### 7.3. Clinical Considerations

As the controversy regarding high drug levels, metabolic ratios, and outcome in terms of adverse reactions demonstrates, the interpretation of the data is challenging. Multiple factors must be taken into account, such as the patients’ smoking behavior, caffeine intake, or co-medication. Whereas the effect of caffeine on metabolism via CYP enzymes is disputed—some sources state that caffeine may lead to an increase of clozapine concentrations [33,242,243] with others claiming caffeine to be merely a substrate of CYP1A2 [108]—smoking habits (which often change during infection, especially respiratory infection) have a relevant effect on the CYP enzyme system, as delineated in Section 2.2. Co-medication is particularly important if the drug inhibits/induces or uses the same CYP enzymes for metabolism [34], such as the antibiotics ciprofloxacin (CYP1A2 and CYP3A4 inhibitor), clarithromycin (CYP3A4 inhibitor), or erythromycin (CYP3A4 inhibitor) [9]. 

Further, inflammation may mask symptoms of drug intoxication if the physiological consequences are similar. In this sense, neuroleptic malignant syndrome typically manifests as changes in blood pressure, leukocytosis, and hyperthermia [244,245]. Other vegetative adverse effects of antipsychotic drugs include orthostatic dysregulation, nasal congestion, or changes in gastrointestinal activity [246], which may also be easily mistaken for a respiratory tract infection or a gastrointestinal disease. A clear attribution of side effects to either elevated drug concentrations or inflammation is therefore often difficult [40]. Interpretation of data is further complicated by the lack of comprehensive literature regarding (1) sex differences in antipsychotic treatment (C/D ratios were found to be generally higher in females than in males according to a study taking 12 antipsychotic drugs into account [17]), and (2) the influence of inflammation on the pharmacokinetics of (a) long-acting antipsychotic drugs (e.g., injectable risperidone, or olanzapine pamoate), and (b) of the elderly considering antipsychotic drugs in general [247]. Particularly the latter consideration is of great relevance, as older individuals often suffer from an increased susceptibility to infections and are more prone to serious ADRs [40,248,249,250]. 

### 7.4. Value of Therapeutic Drug Monitoring during Inflammation—Clinical Recommendations

As the topic ‘TDM of antipsychotic drugs upon inflammatory processes’ is clearly in its infancy, more studies in this direction are urgently warranted to improve psychiatric pharmacotherapy. Especially large-scale prospective studies that systematically assess drug levels (including their respective C/D and metabolic ratios), inflammatory markers, and the clinical consequences, i.e., the presence of any infection-associated ADRs, are required. Regarding the laboratory inflammation markers, CRP (which is mainly expressed under transcriptional control of IL-6 [40,251] with contribution of IL-1 [238,252,253]) was suggested to be appropriate, as it rises and declines rapidly following infection or tissue injury and its subsequent resolution [40,236]. Age, smoking status, weight, lipid levels, and blood pressure were reported to modulate CRP levels. However, keeping the immense increase during an APR in mind (from baseline levels of less than 1 mg/L to 500 mg/L within 24–72 h following severe tissue injury) [238,254,255,256], these factors may be of limited clinical relevance. Further, baseline levels are free of diurnal variations and, according to Pfuhlmann et al. [237], independent of sex. Although a positive correlation between CRP and the leukocyte/white blood cell (WBC) count has previously been reported [257], Scherf-Clavel et al. discourage the use of WBC count as a measure of inflammation [238]. In their study sample, CRP levels and WBC counts were not significantly associated, and—in contrast to CRP—none of the studied antipsychotic drugs seemed to correlate with the number of leukocytes. In fact, the authors emphasize that some bacterial infections (such as dengue or typhoid fever [258,259,260]) are known to cause neutropenia, leading to the assumption that increased leukocyte numbers are a suggestive but not a definitive indicator of infection. Support comes from Leung et al. who state that increased CRP levels are presumably the only sufficient clinical marker of infection, especially in the case of clozapine due to its ability to suppress the WBC count [233]. However, CRP also has its drawbacks as some studies have reported CRP levels to remain stable or to be only moderately elevated upon particular viral infections [261,262]; hence, the optimal marker for inflammation remains yet to be established.

In all of the above-mentioned studies, the authors recommended utmost caution if antipsychotic drugs are taken during an acute inflammation due to the increased risk of intoxication. Drug levels of clozapine are usually increased by three- to fivefold but can also reach concentrations of up to tenfold compared to baseline [40,234,238]. In the study by Hefner et al., a 100% increase of clozapine and risperidone levels (including its active metabolite) were found for CRP values ≥ 25.5 mg/L and ≥37.5 mg/L, respectively. The authors concluded that a fivefold increase of CRP may lead to a twofold increase in clozapine levels [40]. However, if no (or only mild) ADRs are present (see Table 2), the medication can be maintained after careful risk-benefit evaluation. In all other cases the dose should be reduced, or the drug paused. With regard to clozapine, de Leon and Diaz advise to halve the dosage until the patient has recovered from respiratory infection, as the same recommendation was made for theophylline, a drug with comparable metabolization characteristics as clozapine [34]. As for clozapine, theophylline clearance decreases upon infection, as demonstrated by a study injecting endotoxins to humans [263,264,265]. However, in a 46-year old non-smoking patient with bacterial pneumonia, halving the clozapine dose resulted in subtherapeutic levels and concomitant psychotic exacerbation [235].

Besides clozapine, TDM is especially relevant for antipsychotic drugs listed as particularly important regarding a regular TDM (recommendation levels 1 and 2, see Table 1) by AGNP [9]. However, since the pharmacokinetics of a drug depend on a myriad of factors and the impact of inflammation has not been fully characterized to date, caution should also be exercised when using other antipsychotic drugs. If many confounding factors (such as smoking cessation, pregnancy, diet, polypharmacotherapy) are present, consequences may differ depending on the type and capacity of drug-metabolizing enzymes and the particular illness. Considering the high interindividual variability of antipsychotic drug levels, measurement of intraindividual drug levels before, during, and after an episode of inflammation is—together with close monitoring for possible ADRs—strongly recommended. Especially during the ongoing COVID-19 pandemic, one must be particularly attentive: as described in the above mentioned case report of Tio et al., simultaneous infection with COVID-19 and bacterial pneumonia led to ataxia, tremor, and clozapine levels of 1814 ng/mL—an increase thrice as high as during previous respiratory infections—in a non-smoking 46-year-old patient [235]. CRP or cytokine values were not measured in the latter article, but COVID-19 has been reported to cause a “cytokine storm”, i.e., a greater increase in cytokines than in most other diseases. In this case, halving the dosage of clozapine is not sufficient; discontinuation of clozapine until the clinical signs of intoxication resolve may be advisable. 

## 8. Conclusions

Both smoking and inflammation modulate the pharmacokinetic properties of certain drugs. Whereas polycyclic aromatic hydrocarbons in cigarette smoke are potent inducers of hepatic CYP isoenzymes, inflammatory substances (especially IL-6, IL-1, and TNFα) inhibit drug metabolism. The primary pharmacokinetic interactions caused by smoking affect drugs that are substrates of CYP1A2, though other isoenzymes including CYP1A1, CYP1B1, CYP2S1, and CYP2E1 may also be affected. The maximum induction of CYP1A2 is already reached after the consumption of 10 cigarettes. Changes in drug metabolism during inflammation depend on the disease and the concomitantly secreted immunomodulatory substances. It is therefore difficult to predict which CYP enzymes will be primarily affected; however, CYP1A2, CYP2B6, CYP2C8, CYP2C9, CYP2C19, and CYP3A4 are most commonly involved. Further, phase II drug-metabolizing enzymes as well as drug transporters may be inhibited during inflammation, resulting not only in an altered drug metabolism but also in changes in a drug’s distribution and elimination.

Alterations in pharmacokinetics may cause drug intoxication with potentially life-threatening symptoms. TDM as a tool to guide dose titration is thus of importance, especially in psychiatry. The proportion of smokers among patients with a mental illness is two to three times higher than within the general population in Western industrialized nations. Among mentally ill patients, those suffering from schizophrenia are particularly likely to smoke. Further, disturbed functioning of the immune system has been implicated in the pathophysiology of schizophrenia, at least in a subgroup of patients. Patients should be closely monitored for changes in (1) smoking behavior, which are to be expected especially before, during, and after inpatient care, and (2) inflammation (either caused by antipsychotic drugs themselves or pathogens, e.g., severe acute respiratory syndrome coronavirus 2 (SARS-CoV-2)), as these factors may alter drug levels, resulting in the occurrence of ADRs or psychotic exacerbation. Clozapine intoxication presenting with symptoms such as myoclonus, drowsiness, neutropenia, or (paralytic) ileus commonly occurs after a reduction of tobacco consumption or during infection, both resulting in a reduced CYP1A2 activity and thus decreased clozapine metabolism. As certain ADRs of clozapine can be life-threatening and drug levels may drastically increase upon a COVID-19 infection, special precautions should be taken during the ongoing COVID-19 pandemic. Smoking cessation or reduced tobacco consumption as well as increased CRP levels were also found to be significantly associated with elevated drug levels of olanzapine (smoking cessation), or risperidone and quetiapine (inflammation), respectively. However, whereas data on the interaction between smoking and plasma levels of several SGAs and TGAs are abundantly available, the effect of inflammation on drug levels is less well-studied and an influence on other antipsychotic drugs cannot be excluded.

Almost all studies strongly recommend TDM. While smoking is most likely to interfere with drug levels of clozapine and olanzapine, drug levels during inflammation should be monitored especially if the drug is rated with recommendation levels 1 or 2 according to AGNP. Because the pharmacokinetics of a specific drug vary between individuals, caution should also be exercised when using other antipsychotic drugs. To date, there is no standard protocol for guiding clinicians in managing dosing during changed smoking behavior or inflammation; however, some individual suggestions have been proposed. In the case of clozapine, halving the dosage until full recovery from a respiratory infection was found to be reasonable, though this may lead to subtherapeutic levels. Due to “cytokine storms”, risk of clozapine toxicity may not be sufficiently managed by halving clozapine dose in patients with an active SARS-CoV-2 infection. Further, physicians and pharmacists should be aware that due to a concomitant increase of oroso-mucoid/α1-acid glycoprotein with a subsequent reduction of the unbound drug fraction, inflammation may not necessarily result in ADRs despite high clozapine levels. In these cases, the medication can be maintained after careful risk-benefit evaluation. Considering the high interindividual variability of antipsychotic drug concentrations, measurement of intraindividual drug levels before, during, and after an episode of inflammation or changes in smoking behavior is—together with close monitoring for possible ADRs—highly recommended. TDM should be continued after the patient’s discharge from inpatient care, as a return to the original smoking behavior is to be expected.

## Figures and Tables

**Table 1 pharmaceuticals-14-00514-t001:** Therapeutic drug monitoring of second- and third-generation antipsychotic drugs.

Antipsychotic Drug (2nd and 3rd Generation)	Enzymes Involved in Drug Metabolism ^1^	Therapeutic Reference Range	Alert Level	t1/2	TDM Level of Recommendation ^2^
Amisulpride	More than 90% is excreted unchanged via the kidney	100–320 ng/mL	640 ng/mL	12–20 h	1
Comment: Some patients may need concentrations above 320 ng/mL to attain sufficient improvement. CL not affected by CYP enzymes.					
Aripiprazole plus dehydro-aripiprazole	CYP2D6, CYP3A4	150–500 ng/mL	1000 ng/mL	60–80 h	2
Comment: Dehydro-aripiprazole concentrations amount to about 45% of the parent drug. Apparent elimination half-life 30–47 days. CAVE: Steady-state will be reached after approximately 14 days.					
Brexpiprazole	CYP3A4, CYP2D6	40–140 ng/mL	280 ng/mL	91 h	3
Comment: CAVE: Steady-state will be reached after approximately 19 days					
Cariprazine	CYP3A4	10–20 ng/mL	40 ng/mL	48–120 h	3
Comment: Active metabolites are N-desmethyl-cariprazine and N,N-di-desmethyl-cariprazine. CAVE: Steady-state will be reached after approximately 21 days.					
Clozapine	CYP1A2, CYP2C19	350–600 ng/mL	1000 ng/mL	12–16 h	1
Comment: CL may be enhanced in smokers due to induction of CYP1A2 and decreased during inflammation. A lower CRP value associated with a 100% increase in drug serum concentration: 25.5 mg/L *. CL/F is twofold higher in Asian than Caucasian patients. For clozapine, t1/2 is prolonged to 30 h in intoxicated patients.					
Lurasidone	CYP3A4	15–40 ng/mL	120 ng/mL	20–40 h	3
Comment: CL affected by food intake (fat content).					
Olanzapine	UGT1A4, CYP1A2	20–80 ng/mL	100 ng/mL	30–60 h	1
Comment: Apparent half-life for olanzapine pamoate 30 days, CL higher in males than in females and elevated in smokers due to induction of CYP1A2.					
Paliperidone	60% is excreted unmetabolized	20–60 ng/mL	120 ng/mL	17–23 h	2
Comment: Apparent half-life for paliperidone palmitate 25–49 days. CL not affected by CYP enzymes.					
Quetiapine	CYP3A4	100–500 ng/mL	1000 ng/mL	6–11 h	2
Comment: When the patient has taken the extended release (ER) formulation in the evening and blood was withdrawn in the morning, expected concentrations are 2-fold higher than trough levels. CL affected by gender and age. Trend for a drug concentration increase during inflammation (less than 15%) *.					
Risperidone plus 9-hydroxy-risperidone	CYP2D6	20–60 ng/mL	120 ng/mL	2–4 h 17–23 h	2
Comment: Adverse reactions correlate with drug concentrations. To avoid neurological adverse reactions, > 40 ng/mL should be targeted only in cases of insufficient or absence of therapeutic response. Apparent half-life for long-acting injection formulation 26 days. CL affected by CYP2D6 and age, potentially decreased during inflammation. A lower CRP value associated with a 100% increase in RIS + OH-RIS serum concentration was detected at CRP ≥ 37.5 mg/L *.					
Sertindole	CYP2D6	50–100 ng/mL	200 ng/mL	55–90 h	2
Comment: Active metabolite dehydro-sertindole (concentration at therapeutic doses 40–60 ng/mL), concentration dependent increase of QT interval by blockade of potassium channels.					
Sulpiride	Not metabolized, renal excretion	200–1000 ng/mL	1000 ng/mL	8–14 h	2
Comment: CL reduced in case of impaired renal function, CL not affected by CYP enzymes.					
Ziprasidone	-	50–200 ng/mL	400 ng/mL	4–8 h	2
Comment: The drug should be taken with a meal, otherwise absorption is reduced and drug concentrations will be lower than expected.					

CL: Clearance; CRP: C-reactive protein; CYP: cytochrome P450; F: bioavailability; t1/2: elimination half-life; UGT: UDP-glucuronosyltransferase; RIS: risperidone; OH-RIS: 9-hydroxy-risperidone. ^1^ When compounds are combined with strong or moderate inhibitors or inducers of listed enzymes, then the compounds’ concentrations in blood will significantly increase or decrease by ≥ 50%. Therefore, only clinically relevant enzymes involved in drug metabolism are listed. ^2^ Level of recommendation to use TDM: Level 1: Strongly recommended, Level 2: Recommended, Level 3: Useful, Level 4: Potentially useful. * Hefner et al. [40]. Besides Hefner et al. [40], this table also displays data from Hiemke et al. [9].

**Table 2 pharmaceuticals-14-00514-t002:** Pharmacodynamic characteristics of second- and third-generation antipsychotic drugs.

Antipsychotic Drug (2nd and 3rd Generation)	In-Label Diagnoses ^1^	Receptor Profile (Main Receptors Responsible for Drug Efficacy) ^1,2^	Classification AZCERT ^3^	Anticholinergic Activity ^4^	Main Symptoms of Intoxication ^1,2,3,4^
Amisulpride	F20, F21, F23, F25	D2 = D3 > D4 antagonism	Conditional risk	Not classified	Sedation, hypotension, EPMS, QTc-prolongation/TdP
Aripiprazole	F20, F21, F23, F25, F30, F31	D2/D3/5-HT1A partial agonism, 5-HT2A antagonism	Possible risk	None	Somnolence, hypertension, tachycardia, dyspepsia, QTc-prolongation/TdP
Brexpiprazole	F20, F21, F23, F25	D2/D3/5-HT1A partial agonism, 5-HT2A antagonism	Not classified	Not classified	Somnolence, hypertension, tachycardia, dyspepsia
Cariprazine	F20, F21, F23, F25	D2/D3/5-HT1A partial agonism, 5-HT2A/5-HT2B antagonism	Not classified	Not classified	Orthostatic syndrome, somnolence, low blood pressure, abnormal heartbeats, abnormal body movements
Clozapine	F20, F21, F23, F25, G20	H1/α1/5-HT2A/5-HT2C/M1/M4/D4 antagonism	Possible risk	High	Central anticholinergic syndrome, delirium, impaired consciousness, coma, convulsions, hypotension, cardiac adverse events (e.g., QTc-prolongation/TdP), circulatory collapse, respiratory insufficiency, pulmonary edema, metabolic acidosis, (paralytic) ileus
Lurasidone	F20, F21, F23, F25	D2/5-HT2A/5-HT7 antagonism, 5-HT1A partial agonism	Possible risk	Not classified	Arrhythmias, QTc-prolongation/TdP, orthostatic hypotension, circulatory collapse, EPMS, obtundation, seizures, dystonic reaction of the head and neck, aspiration
Olanzapine	F20, F21, F23, F25, F30, F31	mAch/5-HT2/D1-5/H1 antagonism	Conditional risk	Moderate	Central anticholinergic syndrome, delirium, impaired consciousness, coma, agitation, EPMS, circulatory collapse, QTc-prolongation/TdP, tachycardia, respiratory depression, circulatory collapse, NMS
Paliperidone	F20, F21, F23, F25	5-HT2A/5-HT2C/5-HT7/D2 antagonism	Possible risk	Not classified	Delirium, impaired consciousness, coma, agitation, EPMS, circulatory collapse, QTc-prolongation/TdP, tachycardia, respiratory depression, circulatory collapse, NMS
Quetiapine	F20, F21, F23, F25, F30, F31, F32, F33, F34, F43.2	H1/5-HT1/5-HT2/D1-3 antagonism	Conditional risk	Low	Delirium, impaired consciousness, coma, agitation, EPMS, circulatory collapse, QTc-prolongation/TdP, tachycardia, respiratory depression, circulatory collapse, NMS
Risperidone	F0, F20, F21, F23, F25, F30, F31	5-HT2A/5-HT2C/5-HT7/D2 antagonism	Conditional risk	None	Delirium, impaired consciousness, coma, agitation, EPMS, circulatory collapse, QTc-prolongation/TdP, tachycardia, respiratory depression, circulatory collapse, NMS
Sertindole	F20, F21, F23, F25	D2/5-HT2 antagonism	Known risk	Not classified	Respiratory depression, QT-prolongation/TdP/cardiac death, EPMS, circulatory collapse
Sulpiride	F20, F21, F23, F25, F31, F32, F33, F34, F43.2	D2/D3 antagonism	Known risk	Not classified	EPMS, impaired consciousness, agitation, coma, hypotension, QT-prolongation/TdP, cardiac death
Ziprasidone	F20, F21, F23, F25	5-HT2A/5-HT2C/D2/D3/H1 antagonism	Conditional risk	None	Delirium, impaired consciousness, coma, agitation, EPMS, circulatory collapse, QTc-prolongation/TdP, tachycardia, respiratory depression, circulatory collapse, NMS

EPMS: extrapyramidal motoric symptoms, NMS: Neuroleptic malignant syndrome, TdP: Torsade de pointes. ^1^ summary of product characteristics, see Table A1 for further information regarding the ICD-10 coding; ^2^ Benkert, Hippius [108]; ^3^
www.crediblemeds.org [109]; Classification AZCERT: Known Risk of TdP—These drugs prolong the QT interval AND are clearly associated with a known risk of TdP, even when taken as recommended. Possible Risk of TdP—These drugs can cause QT prolongation BUT currently lack evidence for a risk of TdP when taken as recommended. Conditional Risk of TdP—These drugs are associated with TdP BUT only under certain conditions of their use (e.g., excessive dose, in patients with conditions such as hypokalemia, or when taken with interacting drugs) OR by creating conditions that facilitate or induce TdP (e.g., by inhibiting metabolism of a QT-prolonging drug or by causing an electrolyte disturbance that induces TdP). ^4^ Hiemke, Eckermann [110].

## Data Availability

Not applicable.

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
