# Peer review of "Therapeutic Drug Monitoring of Second- and Third-Generation Antipsychotic Drugs—Influence of Smoking Behavior and Inflammation on Pharmacokinetics"

_pharmaceuticals, 2021, doi:10.3390/ph14060514_

Round 1

Reviewer 1 Report

The review generalizes the information on the influence of smoking and inflammation on pharmacokinetics of antipsychotic drugs. The review collects numerous references, very well structured and written. In my opinion, the review can be interesting and useful not only for clinicians but for specialists in bioanalysis and new drugs development.

Nevertheless, I'd like to raise some questions and points to discuss.

1) Smoking is known to affect the microbiome. The latter is known both to contribute in drugs pharmacokinetics and and to influence on disease course or even to modulate it. My suggestion or even a question is whether it is possible to discuss the role of microbiome in connection with the review's topic.

2) Table 1 (page 10): The column "Comments" is spoiling the Table's look. I'd recommend to increase its width or place the text in a merged line below the information on a corresponding drug.

3) Pages 16-17: There is a paragraph starting on the page 16 and continued on the page 17, where some analytical methods for TDM are discussed. This paragraph must be corrected or, better, removed as it contains some wrong statements (e.g., description of MS/MS as two mass spectrometers, or separation of ions by magnetic fields). At least, consult some specialists in mass spectrometry to correct the text.

4) Page 23, Table A1: Some titles of the classifications are different from those found online (please visit https://icdlist.com/icd-10/index/mental-behavioural-disorders). As I am not a specialist in this area, I'd like to recommend additional checking of the classification.

5) My last question is concerned with the topic of the review in general. Any drug, before being admitted for the use, is tested on large cohorts. Within the cohorts, there were, I suppose, smoking people, so, the data on the pharmacokinetics of a drug in smokers are averaged with those of non-smokers. As such, why such a big difference was observed in studies which are considered in the review?

Reviewer 2 Report

Manuscript 1226224 is a narative review that provides comprehensive review of relevant literature on very important topic that is clinically relevant in psychiatry. Both inflammation and smoking can influence a drug’s pharmacokinetic properties, thus this manuscript may be very important to spread awareness regarding TDM, as a tool to optimize drug safety and provide patient-tailored treatment, particularly in the context of antipsychotic drugs (such as clozapine and olanzapine), inflammation, and smoking behavior. This manuscript is written in a clear manner, it is a comprehensive review of relevant published data on this important topic, with clinical recommendations and concise and appropriate conclusion.

Author Response

We would kindly like to thank Reviewer #2 for the positive feedback on our review. We are pleased to hear that the Reviewer considers our manuscript to be a relevant and useful contribution and that we have succeeded in providing a comprehensive overview of this very complex matter.

Reviewer 3 Report

This manuscript is an interesting review of the most relevant literature regarding the influence of smoking behavior and Inflammation in the pharmacokinetic of Second and Third generation antipsychotics, with special emphasis on the Therapeutic Drug Monitoring of plasma/blood drugs levels. The authors have made an elegant and comprehensive description of those metabolic and pharmacokinetic parameters to keep in mind in psychiatric drug prescription. 

Author Response

We would kindly like to thank Reviewer #3 for the positive feedback on our review. We are pleased to hear that the Reviewer considers our manuscript to be a relevant and useful contribution and that we have succeeded in providing a comprehensive overview of this very complex matter.